# Shelf-Life Extension of Wood Apple Beverages Maintaining Consumption-Safe Parameters and Sensory Qualities

**Md. Shakir Moazzem, Md. Belal Hossain Sikder * and Wahidu Zzaman**

Department of Food Engineering and Tea Technology, Shahjalal University of Science and Technology, Sylhet 3114, Bangladesh; shakir.sust@gmail.com (M.S.M.); wahidanft@yahoo.com (W.Z.)

*   Correspondence: belalustc@yahoo.com; Tel.: +880-1911-212427

**Abstract:** An investigation was carried out to extend the shelf life of wood apple beverages by up to 50 days from its natural shelf life of 8–12 h. A wood apple beverage was prepared using freeze-dried wood apple powder. Four samples were developed by pasteurizing the beverage at 85 °C for 10 min and treatment with a combination of 50 ppm and 100 ppm of potassium metabisulphite, citric acid, ascorbic acid, and sodium benzoate. Replications and controls were properly maintained. The total soluble solids (°Brix), pH, titrable acidity, ascorbic acid content, microbial growth and sensory attributes of the prepared juice samples were evaluated at an interval of 10 days over a storage period of 50 days. TSS was found to increase (16.30–18.25°Brix) with storage period, while pH (5.43–4.10), titratable acidity (0.67–0.08%), and ascorbic acid content (4.65–1.01 mg/100 mL) decreased with time. The microbiological analysis showed little or no growth for samples treated with a combination of 50 ppm potassium metabisulphite, citric acid, ascorbic acid and sodium benzoate up to 50 days. Consumer acceptability of the beverage was found to be satisfactory. Thus, shelf life of wood apple beverage was extended to 50 days satisfactorily, ensuring consumption-safe parameters and satisfactory sensory qualities.

**Keywords:** shelf-life extension; sensory qualities; pasteurization; product development; consumption-safe parameters

## 1. Introduction

The wood apple (*Aegle marmelos* L. Correa) is an important indigenous fruit of the Indian subcontinent, belonging to the family Rutaceae. It is native of the Indo-Malayan region and has been known in India since prehistoric times [1]. It grows in the dry forests, hills, and plains of India, Myanmar Bangladesh, and Pakistan [2]. The importance of wood apple lies in its medicinal and curative properties [3]. It is one of the most useful medicinal fruits of the Indian subcontinent [4]. The wood apple is considered to be a natural source of anti-oxidants due to its potential radical scavenging activity of various phytochemicals [5]. It also has hypoglycemic, antitumor, larvicidal, antimicrobial, and hepatoprotective activity [6]. It has anti-diabetic and antioxidant potential in terms of reducing levels of blood glucose and malondialdehyde [7]. Compounds purified from wood apples have been proven to be biologically active against several major diseases including cancer and diabetes. [8]. In Ayurveda, it is used to cure problems related to the heart, stomach, and intestine, chronic constipation, dysentery, some forms of indigestion, typhoid, debility, fever, hemorrhoids, hypochondria, melancholia, and heart palpitations [9].

The wood apple is one of the most nutritious fruits of the Indian subcontinent [1]. It contains many vitamins, such as vitamin A, vitamin C, thiamine, riboflavin, and niacin, and minerals, such as

calcium and phosphorus [4]. The edible pulp of 100 g of wood apple fruit contains 61.0 g of moisture, 1.6 g of protein, 0.2 g of fat, 1.9 g of minerals, 80.0 mg of calcium, 52.0 mg of phosphorus, 0.5 mg of iron, 55 μg of carotene, 0.12 mg of thiamine, 1.19 mg of riboflavin, 1.0 mg of niacin, 8 mg of vitamin C, 610 mg of potassium, and 0.20 mg of copper [10]. Wood apple is very rich in vitamins, amino acids, and minerals in comparison with other fruits [11] and can contribute significantly to the daily nutrient needs of the individual. In addition, it can be used advantageously to supplement deficiencies of other foods [12]. The ripe fruit of the wood apple is sweet, aromatic, nutritious, and very palatable and is highly esteemed and eaten by all classes of people in India [11]. However, people prefer adding it to other products to consuming it on its own [2]. People of the Indian subcontinent usually consume wood apple as a beverage after blending it with other ingredients [13].

Fruit-based drinks are far superior to many synthetic preparations and are being replaced by fruit beverages. Fruit beverages are easily digestible, highly refreshing, thirst quenching, appetizing, and nutritionally far superior to many synthetic and aerated drinks [14]. They can help one to meet the daily requirement of fruits and vegetables in one's diet [15]. There are many pathways for the deterioration of fruit beverages. However, many effective preservation methods can prevent spoilage. The preservation of fruit beverages can be achieved through processing and the use of preservatives. Fruit beverages tend to deteriorate and spoil easily if they do not undergo heat treatment and treatment with preservatives. The most common method of inactivating microorganisms and enzymes for increasing the shelf life of fruit juices is by thermal processing such as pasteurization. Chemical preservatives are also used to prolong the shelf-life of fruit juice and beverages and inhibit microbial growth [16]. The inhibitory action of preservatives is due to their interfering with the mechanism of cell division, the permeability of cell membranes, and the activity of enzymes. Chemical preservatives such as potassium metabisulphite, sodium benzoate, citric acid, ascorbic acid, and their mixtures have been used widely in fruit beverages and are generally recognized as safe (GRAS) [17]. However, excessive consumption of these preservatives may be hazardous to human health [18]. In order to ensure the safety of fruit beverages for human consumption, chemical preservatives have to be used within acceptable limits. The improper use of preservatives exceeding the acceptable limit can be a potential risk for human health. In the last decades, food safety has become very important for governments, producers of food products, as well as consumers [19]. According to the 2018 Food Safety (Food Hygiene) Regulations of the Bangladesh Food Safety Authority (BFSA), the maximum limits of potassium metabisulphite and sodium benzoate in the industrial production of fruit beverages are 1000 and 600 ppm, respectively, whereas no limit has been set for using citric acid and ascorbic acid in fruit beverages because of their abundance in citrus fruits [20]. The 2011 Food Safety and Standards (Food Product Standards and Food Additives) Regulation of the Food Safety and Standards Authority of India (FSSA) specifies that the maximum limits of potassium metabisulphite and sodium benzoate for the industrial production of fruit juices and beverages are 700 and 600 ppm, respectively [21]. It should be noted that potassium metabisulphite, sodium benzoate, citric acid, and ascorbic acid are considered GRAS by the FAO [22] and the FDA [23].

The influence of chemical preservatives on the quality attributes of orange juice was investigated by Stephen et al. It was possible to store orange juice effectively for three weeks using 0.03% sodium benzoate, sodium metabisulphite, potassium sorbate, or its combinations of preservatives [24]. The effect of chemical preservatives on strawberry juice was investigated by Ayub et al. Pasteurization and treatment with 20% sucrose, 0.1% sodium benzoate, and 0.1% potassium sorbate prolonged the shelf-life of strawberry juice by up to three months [25]. The clarification, preservation, and shelf life evaluation of cashew apple juice were investigated by Talasila et al. It was possible to prolong its shelf life by up to 90 days by treatment with 0.1 g/L citric acid and benzoic acid [26]. Various factors impacting wood apple beverage production were investigated by Minh [27]. The comparative effect of crude and commercial enzymes on the juice recovery from wood apple (*Aegle marmelos* Correa) using principal component analysis was investigated by Singh et al. The effects of incubation time, incubation temperature, and crude enzyme concentration on the yield, viscosity, and clarity of the juice obtained

from wood apple fruit pulp were observed. It was concluded that wood apple juice yield, viscosity, and clarity are functions of enzymatic hydrolysis conditions [28]. The effect of thermal processing on the preparation of ready-to-serve wood apple beverage blended with aonla was studied by Rathod et al. It was observed that pasteurization by thermal processing (90 °C for 25 s) was effective for inactivating microbial flora [14]. The effects of the processing quality and storage stability of functional beverages prepared from aloe vera blended with wood apple fruit were studied by Sasikumar. It was concluded that a formulation of functional beverages can satisfy consumers' acceptance [2]. It was also indicated by Tiwari and Deen that wood apple and aloe vera can be utilized for a valuable RTS (ready-to-serve) beverage and is acceptable to consumers in terms of taste, color, flavor, and medicinal properties [29]. The processing and storage stability of wood apple squash for nutritional security was investigated by Mandal et al. It was concluded that it may play an important role in food and nutritional security [3]. The wild wood apple fruit was studied by Kenghe et al. in terms of value addition. Squash was prepared from this fruit by adjusting the TSS (total soluble solids) of the pulp [30]. The preparation and shelf-life of mixed juice based on wood apple and papaya were investigated by Chowdhury et al. It was concluded that potassium metabisulphate (KMS) was effective for the prevention of spoilage against microbial growth in bottled mixed juices [31]. Jam and fruit bars with wood apple were developed by Vidhya and Narain and were safe and fit for consumption for up to 90 days [32]. The effect of different pre-treatments on the physicochemical properties, organoleptic quality, and shelf life of wood apple candy was investigated by Kumar et al. for up to 4 months [1]. Candy, jam, murabba, and chutney using wood apple by incorporation of various herbs (ginger, cardamom, and rose extract in wood apple) were developed by Srivastava et al., and the acceptability and nutritive value of the products were assessed [33].

Currently, processed products based on wood apple are becoming popular because of its rich nutrient profile [27]. Wood apple has an excellent aroma that is not removed during processing. Therefore, there is untapped potential for processing wood apple into various products [34]. It also has great potential for commercial processing and can be processed into various products such as beverages, preserves, candy, squash, toffee, slabs, pulp powder, and nectar [35].

It is evident that wood apple beverages are highly nutritious, with genuine medicinal value, and they hold great potential in the developing market of beverages to become a popular, high-quality beverage of commercial interest. Some attempts have been made very recently to develop value-added products such as jam, fruit bars, squash, murabba, chutney, and candy. However, no serious effort has been made to extend the shelf-life of wood apple beverages to the best of our knowledge. Therefore, it would be of great significance to extend the shelf-life of such highly nutritious beverages to make them available for longer periods of time. The present investigation was carried out to extend the shelf-life of a wood apple beverage by up to 50 days from its natural shelf-life of 8–12 h while maintaining physicochemical, microbial, and sensory attributes. It was also the aim of this study to establish a set of formulae for the commercial production of shelf-stable wood apple beverages on an industrial scale.

## 2. Materials and Methods

### 2.1. Materials and Chemical Reagents

Healthy, fresh, and fully ripe wood apple fruits, with a rich sweet aroma and without any visual defects, were purchased from the Swapno super shop of Subidbazar, Sylhet, Bangladesh. All the chemicals used in this experiment were purchased from Hi Media®, India, and were of analytical grade. The Laboratory of Dept. of Food Engineering and Tea Technology provided this investigation with potassium metabisulphite, sodium benzoate, ascorbic acid, citric acid, plate counting agar for total viable bacterial count, potato dextrose agar for total fungal count, McConkey agar for coliform counts, refined sugar, distilled water, and other chemicals and reagents. Experiments mentioned in

this study were conducted between June and August of 2018 in Sylhet, Bangladesh. Room temperature is usually around 28–32 °C.

## 2.2. Preparation of Freeze-Dried Wood Apple Powder from Raw Wood Apple Pulp

Wood apple fruits were taken to the laboratory of Dept. of Food Engineering and Tea Technology and washed in tap water to wash out adhering dirt and dust particles. Fruits were cut in slices by a wood cutter. Bark and seeds were removed, and the pulp was scooped out with the help of a stainless steel spoon. The extracted pulp was then kneaded and heated at 75 °C for 2 min. A No. 20 mesh stainless steel sieve (0.841 mm/841 microns) was used to sieve the pulp and remove the seeds and fibers. A TelStar LyoQuest freeze drier (Telstar Technologies, Barcelona, Spain) was used to make freeze-dried wood apple powder from the pulp.

## 2.3. Preparation of Beverage

A total of 2 kg of freeze-dried powder was obtained from 6 kg of ripe wood apple fruits. A total of 1.5 L of wood apple beverage was produced in 3 batches by mixing a total of 150 g of freeze-dried wood apple powder and a total of 150 g of sugar with a total of 1.5 L of distilled water. In each batch, 500 mL of wood apple beverage was made by mixing 50 g of freeze-dried wood apple powder and 50 g of sugar with 500 mL of distilled water. Pasteurization of the prepared wood apple beverage was done by heating at 85 °C for 10 min. It was then cooled down at a room temperature of 32 °C for 15 min. The main beverage sample was then taken into 12 different glass bottles and treated with a combination of 50 ppm and 100 ppm of potassium metabisulfite (KMS), citric acid (CA), ascorbic acid (AA), and sodium benzoate (SB). The glass bottles were then corked tightly and wrapped with aluminum foil in an aseptic condition. Flow chart showing procedure for preparing wood apple beverage samples is shown in Figure 1.

## 2.4. Labeling of Samples

The labelling of samples is given in Table 1. Control samples, i.e., samples not treated with any preservatives, were labeled as Sample A. The samples treated with 100 ppm KMS (potassium metabisulphite) + 100 ppm CA (citric acid) were labeled as Sample B. The samples treated with 100 ppm of AA (ascorbic acid) + 100 ppm SB (sodium benzoate) were labeled as Sample C. The samples treated with 50 ppm KMS (potassium metabisulphite) + 50 ppm CA (citric acid) + 50 ppm of AA (ascorbic acid) + 50 ppm SB (sodium benzoate) were labeled as Sample D. The beverage samples treated with these preservatives were stored in a room temperature of 28–32 °C for further evaluation.

**Table 1.** Labeling of samples showing doseage of heat treatment and concentrations of preservatives.

| Sample Labels | Heat Treatment, i.e., Pasteurization | Potassium Metabisulphite (KMS) | Sodium Benzoate (SB) | Citric Acid (CA) | Ascorbic Acid (AA) |
|---|---|---|---|---|---|
| Sample A (Control) | 85 °C for 10 min | - | - | - | - |
| Sample B | 85 °C for 10 min | 100 ppm | - | 100 ppm | - |
| Sample C | 85 °C for 10 min | - | 100 ppm | - | 100 ppm |
| Sample D | 85 °C for 10 min | 50 ppm | 50 ppm | 50 ppm | 50 ppm |

- designated preservative has not been used in any form or concentration.

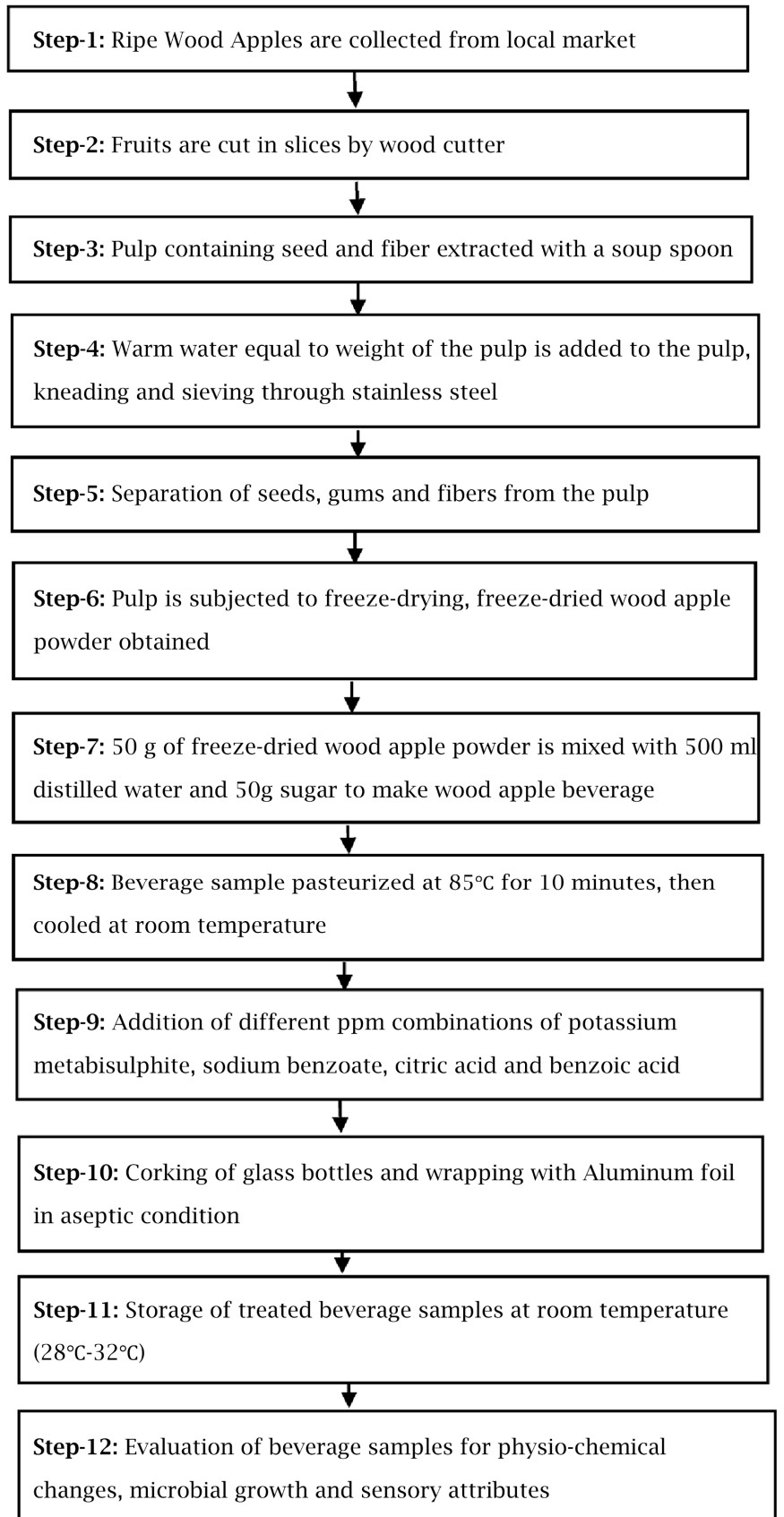

**Figure 1.** Flow chart showing procedure for preparing wood apple beverage samples.

## 2.5. Determination of Physicochemical Properties

TSS (°Brix) was determined by the method of AOAC (2012) by using an Abbe Hand Refractometer (Erma, Tokyo, Japan) [36]. A sample replaceable prism was inserted in the refractometer. It was then held against a light source. The reading was regarded as the total soluble solids of the samples in °Brix. pH was determined by the method of AOAC (2012) [36]. pH values of the beverage samples were measured at room temperature with a digital glass electrode pH meter (Model 744, Metrohm, Herisau, Switzerland), which was calibrated prior to sample pH measurement using standard buffer solutions of pH 4 and 7. About 10 mL of sample was measured into a beaker, and the pH meter was dipped into the beaker to measure the pH of the sample. The pH values were then recorded. Titratable acidity (as citric acid) was estimated by the method described by Ranganna [37]. Titratable acidity was determined by dissolving a known weight of sample in distilled water and titration against 0.1 N NaOH using phenopthalein as an indicator. Ascorbic acid content (as vitamin C) was estimated by a method described by Ranganna [37]. Ascorbic acid content was determined by a direct colorimetric method using 2,6-dichlorophenol indophenol as a decolorizing agent by ascorbic acid in the sample extract and in a standard ascorbic acid solution.

## 2.6. Microbial Examination

Samples were analyzed for total bacterial count, total fungal count, and coliform count according to the procedure described by APHA [38]. The serial dilution and plating method was followed to enumerate the microbial load of wood apple beverage samples. Enumeration of total bacterial count, total fungal count, and coliform count was done by using plate counting agar, potato dextrose agar, and McConkey agar, respectively. One milliliter of juice from each sample was taken into a test tube containing 9 mL of sterile water and homogenized by shaking to make the $10^{-1}$ dilution. Serial dilutions of $10^{-2}$, $10^{-3}$, $10^{-4}$, $10^{-5}$, and $10^{-6}$ were then prepared from it. The plate counting agar plates (with $10^{-6}$ dilutions) and potato dextrose agar plates (with $10^{-4}$) were incubated at 32 °C for 48 h for total bacterial count and total fungal count, respectively. McConkey agar plates (with $10^{-2}$ dilutions) were incubated at 37 °C for 24 h for coliform count. The colony enumeration was done using a digital colony counter, and values were expressed as colony forming units/mL (cfu/mL) of the sample. The following formula was used to calculate the number of colony forming units per mL (cfu/mL) after the incubation time:

$$\text{cfu} = \frac{\text{Number of colony}}{\text{volume of sample added} \times \text{dilution factor}}.$$

## 2.7. Sensory Evaluation

Data on sensory evaluation of this study was obtained according to the recommendations made by the Society of Sensory Professionals [39]. The consumers' acceptance of the juice was evaluated by 150 consumers (naïve panelists). The sensory panel comprised of randomly selected people from Sylhet, Bangladesh with ages ranging from 8 to 67 years. Samples were presented in 200 mL glass bottles, and the panelists were asked to evaluate the samples for appearance, aroma, taste, and overall acceptability using a nine-point hedonic scale, varying from "dislike extremely" (Score 1) to "like extremely" (Score 9). Sensory booths were set up in the laboratory of Food Engineering and Tea Technology with proper ventilation, a neutral background, proper lighting, good internet connectivity, minimal traffic, and no distractions, noise, or odors. Samples were prescreened to ensure that they represent the important parameters being tested and did not contribute to unnecessary bias. The serving bottles were masked with aluminum foil with no suggestive information so that respondents could not identify samples unintentionally. Brand new palate cleaners and glass bottles were used. Bias was minimized by serving the samples to the panelists in masked 200 mL glass bottles at a room temperature of 28–32 °C. All samples were held for at least 3 min for proper sensation so that the panelists could distinguish

between nuanced differences. Five minutes of break time was maintained after each evaluation to eliminate fatigue and exhaustion.

## 2.8. Interval at Evaluation Samples

Physicochemical properties (TSS, pH, titratable acidity, and ascorbic acid content), microbial growth (total bacterial count, total fungal count, and coliform count), and sensory attributes (color, flavor, taste, and overall acceptability) were evaluated at an interval of 10 days over a period of 50 days.

## 2.9. Statistical Analysis

Data were analyzed using SPSS software (SPSS Inc., Chicago, IL, USA), version 25 for Windows. Results of TSS (°Brix), pH, titratable acidity (%) and ascorbic acid content (mg/100 mL) were reported as mean $\pm$ standard deviation (SD) for three (3) replicates, *n* = 3. Results of sensory evaluation for appearance, aroma, taste, and overall acceptability were reported as mean $\pm$ standard deviation (SD) of 9-point hedonic scale ratings given by 150 consumers, *n* = 150. One-way analysis of variance (ANOVA) followed by Duncan's multiple range test (DMRT) (multiple comparison post-hoc test) was used to analyze the statistical difference. Differences with *p*-values < 0.05 were considered statistically significant.

## 3. Results and Discussion

### 3.1. Changes in Physicochemical Properties

TSS (°Brix) of the samples gradually increased from the first day (16.30°Brix to 16.50°Brix) to the end of storage (16.78°Brix to 18.25°) throughout the storage period (Table A1). Sample D showed better retention of TSS in comparison with other samples. The changes in TSS (°Brix) of the beverages are shown in Figure 2. The increase in TSS during storage may be attributed to a conversion of polysaccharides into simple sugars and a degradation of pectic substances in soluble solids [40]. This result was also found in reports of Singh et al. [40], Minh [27], Tiwari and Deen [29], Mandal et al. [3], Rathod et al. [14], and Chowdhury et al. [31]. Similar observations were also reported by Kumar et al. [1] for wood apple candy.

pH of the samples gradually decreased from the first day (5.43–4.88) to the end of storage (4.97–1.30) throughout the storage period (Table A1). The higher retention of pH by Sample D in comparison with other samples is depicted in Figure 3. A decline in pH during storage was observed, which may be due to the action of citric and ascorbic acid on the sugar and protein component of the product [10]. This result is in accordance with those of Rathod et al. [14] and of Minh [27]. Such a decrease in pH over the storage period was also reported by Yang et al. [18] for orange juice and by Krishnakumar et al. [41] and Khare et al. [42] for sugarcane juice. However, a better retention of pH could be obtained in this study, which indicates the substance is safe for human consumption.

Titratable acidity (%) of the samples gradually decreased from the first day (0.69–0.43%) to the end of storage (0.43–0.08%) throughout the storage period (Table A1). The decrease in acidity was rapid in Sample A, which was not treated with any preservatives. A decrease in acidity during storage might be due to chemical interactions between organic constituents and enzymatic reactions [40]. It might also be due to the conversion of acids into salt sand sugars by enzymes, particularly, Invertase [9]. This result is in agreement with those of Chowdhury et al. [31], Singh et al. [40], and Verma and Gehlot [43]. Similar observations were reported by Dhaka et al. for kinnow juice [44] and Paul and Ghosh for pomegranate juice [45]. The higher retention of titratable acidity (%) by Sample D in comparison with other samples is shown in Figure 4.

Ascorbic acid content of wood apple beverage samples gradually decreased from the first day (4.65–4.43 mg/100 mL) to the end of storage (3.01–1.01 mg/100 mL) throughout the storage period (Table A1). This reduction may be due to the oxidation of ascorbic acid in to dehydrate ascorbic acid by

oxygen [3]. Ascorbic acid is sensitive to oxygen, light, and heat and is easily oxidized in the presence of oxygen by both enzymatic and non-enzymatic catalysts [14]. This finding was also found in the reports of Tiwari and Deen [29], Rathod et al. [14], Mandal et al. [3], and Chowdhury et al. [31]. Similar observations were reported by Bhardwaj and Mukherjee [46] on kinnow juice blends, Talasila et al. [26] on cashew apple juice, and Khare et al. [42] on sugarcane juice. The higher retention of ascorbic acid content (mg/100 mL) by Sample D in comparison with other samples is depicted in Figure 5.

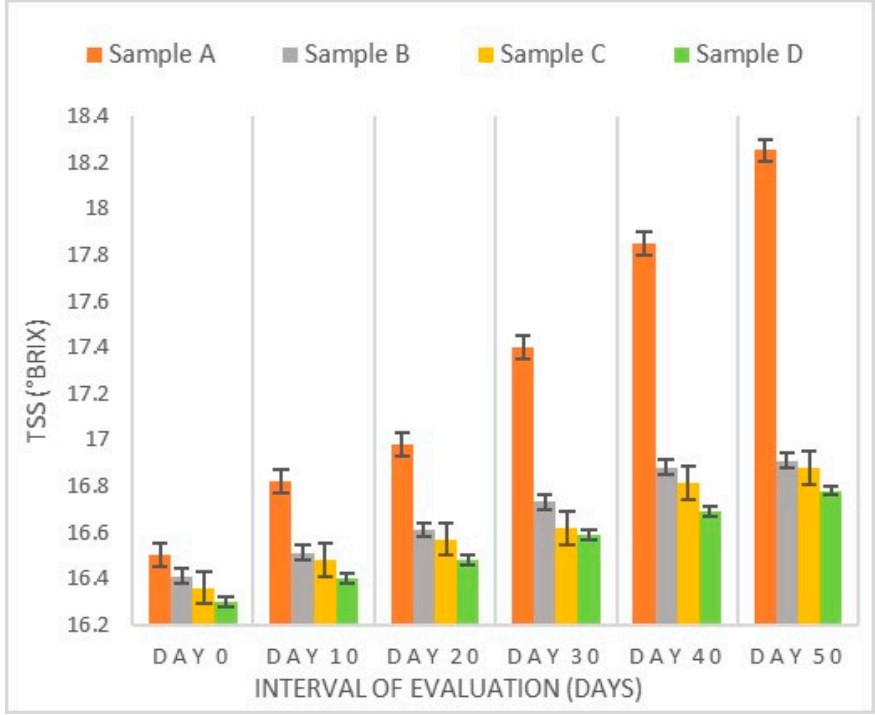

**Figure 2.** TSS (°Brix) evaluation of samples.

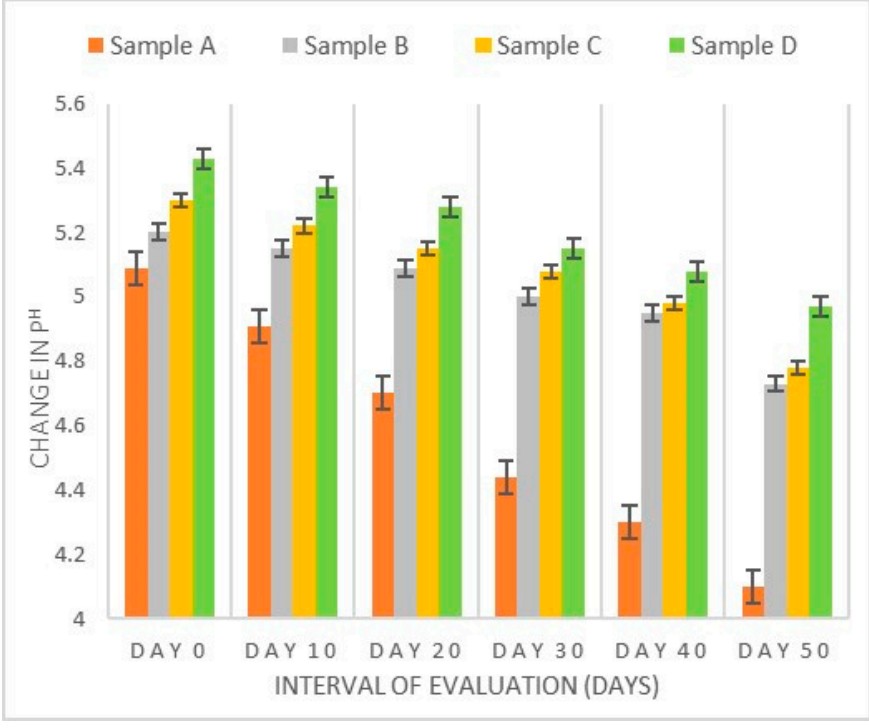

**Figure 3.** pH evaluation of samples.

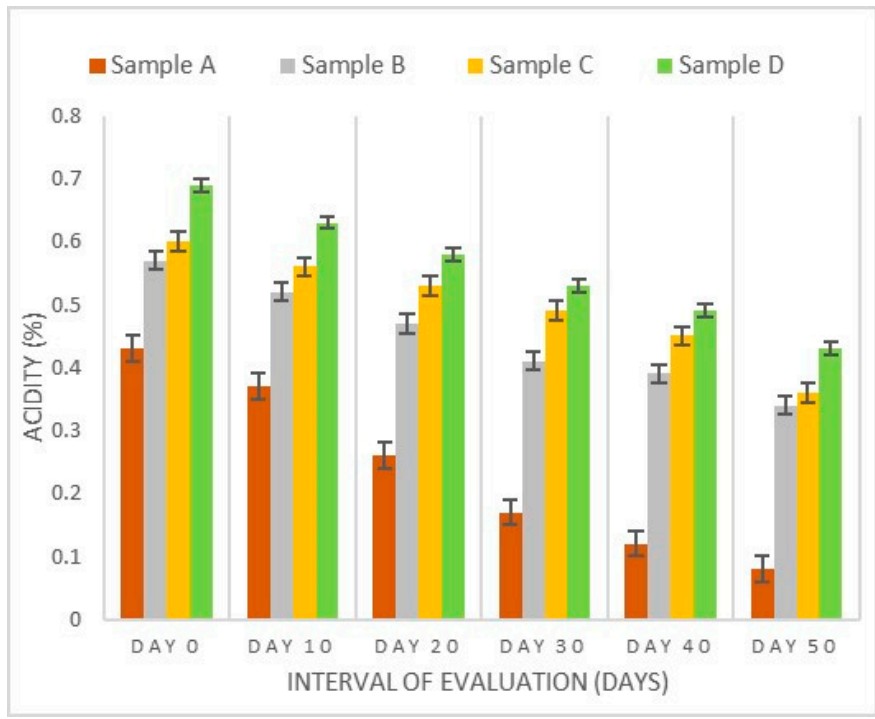

**Figure 4.** Acidity (%) evaluation of samples.

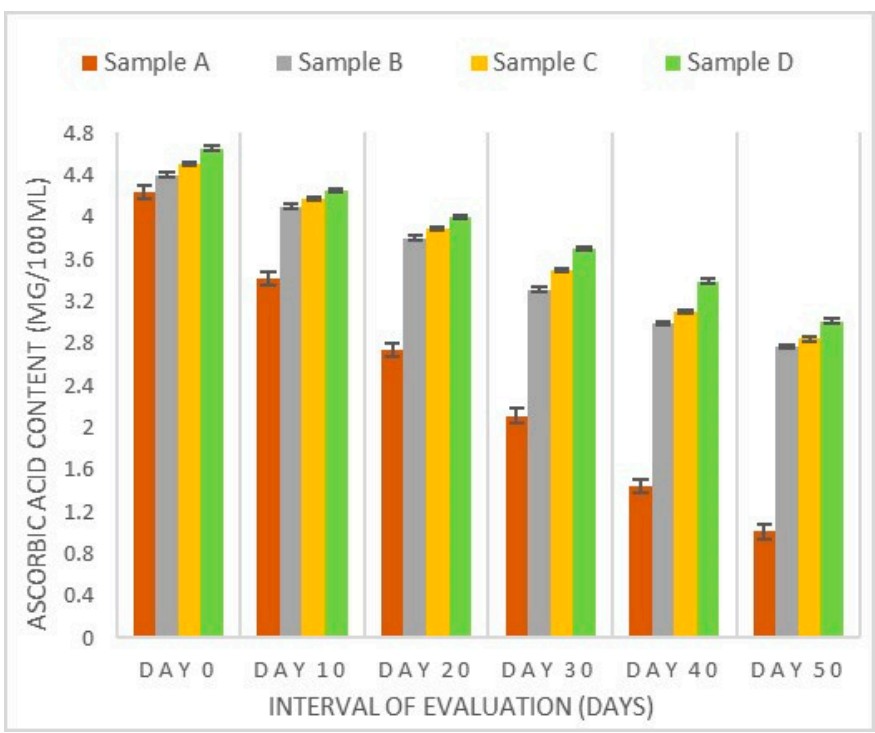

**Figure 5.** Ascorbic acid evaluation of samples.

*3.2. Changes in Microbial Load*

The microbial evaluation of wood apple beverage is given in Table 2. The bacterial load of Sample A (control samples) increased considerably from the day of preparation ($1.09 \times 10^7$ cfu/mL at Day 0) to the end of storage ($2.43 \times 10^7$ cfu/mL at Day 50). This might be due to the fact that Sample A was not treated with any preservatives. As for treated juice samples, bacterial load was observed to be less than 10 cfu/mL at the day of preparation (Day 0). Heat and the action of preservatives destroyed most

of the microbes. The presence of bacterial load was noticed at the end period of storage, notably at Day 40 (under $3 \times 10^5$ cfu/mL) and Day 50 (under $10 \times 10^5$ cfu/mL) in Sample C. Sample B developed bacterial load at Day 50 (under $2 \times 10^5$ cfu/mL). Sample D developed no bacterial colony during the storage of 50 days, but the presence of bacterial load was noticed at Day 60 ($3.78 \times 10^5$ cfu/mL). Total fungal count (yeast and mold count) of control samples (Sample A) increased considerably from the day of preparation (no growth at Day 0) to the end of storage period ($7.0 \times 10^5$ cfu/mL at Day 50). As for treated juice samples, no fungal load was observed at Day 0. The presence of a few fungal loads was noticed at the end period of storage in Sample B (under $4 \times 10^4$ cfu/mL at Day 50) and Sample C (under $2 \times 10^4$ cfu/mL at Day 50). Sample D developed no fungal growth over the storage period of 50 days, but fungal growth was observed at Day 60 ($2.83 \times 10^4$ cfu/mL). No growth of indicative organisms such as coliforms was observed in the samples over the storage period. Growth of coliforms was inhibited by heat treatment and the inhibitory action of preservatives.

Thus, pasteurized wood apple beverage samples treated with a combination of 50 ppm and 100 ppm of preservatives were safe to drink for up to 50 days. The results of microbial evaluation found in this study is in accordance with the reports of Sasikumar [2] and Chowdhury et al. [31] and are in agreement with the reports of Akinola et al. [24], Ayub et al. [25], Bhardwaj and Mukherjee [36], Khare et al. [42], Dhaka et al. [44] and Chatterjee et al. [47].

**Table 2.** Evaluation of total viable count of samples over storage period.

| Type | Sample No. | Time of Storage (Days) | | | | | | |
|---|---|---|---|---|---|---|---|---|
| | | Day 0 (cfu/mL) | Day 10 (cfu/mL) | Day 20 (cfu/mL) | Day 30 (cfu/mL) | Day 40 (cfu/mL) | Day 50 (cfu/mL) | Day 60 (cfu/mL) |
| Total bacterial count | Sample A | $1.09 \times 10^7$ | $1.52 \times 10^7$ | $1.71 \times 10^7$ | $1.88 \times 10^7$ | $2.00 \times 10^7$ | $2.43 \times 10^7$ | $3.53 \times 10^7$ |
| | Sample B | Nil | Nil | Nil | Nil | Nil | $\leq 2.00 \times 10^5$ | $3.26 \times 10^6$ |
| | Sample C | Nil | Nil | Nil | Nil | $\leq 3.00 \times 10^5$ | $\leq 1.00 \times 10^6$ | $2.63 \times 10^6$ |
| | Sample D | Nil | Nil | Nil | Nil | Nil | Nil | $3.78 \times 10^5$ |
| Total yeast and Mold count | Sample A | Nil | $3.30 \times 10^5$ | $4.00 \times 10^5$ | $4.90 \times 10^5$ | $6.00 \times 10^5$ | $7.00 \times 10^5$ | $1.50 \times 10^6$ |
| | Sample B | Nil | Nil | Nil | Nil | Nil | $\leq 4.00 \times 10^4$ | $9.00 \times 10^4$ |
| | Sample C | Nil | Nil | Nil | Nil | Nil | $\leq 2.00 \times 10^4$ | $7.00 \times 10^4$ |
| | Sample D | Nil | Nil | Nil | Nil | Nil | Nil | $2.83 \times 10^4$ |
| Total coliform count | Sample A | Nil | Nil | Nil | Nil | Nil | Nil | Nil |
| | Sample B | Nil | Nil | Nil | Nil | Nil | Nil | Nil |
| | Sample C | Nil | Nil | Nil | Nil | Nil | Nil | Nil |
| | Sample D | Nil | Nil | Nil | Nil | Nil | Nil | Nil |

Nil = No microbial growth.

*3.3. Changes in Sensory Attributes*

Results on sensory evaluation are presented in Table 3. The DMRT test that followed ANOVA revealed that there was a significant difference (with significance level of 0.05) among samples in their sensory attributes. Sample D was significantly "better" than other samples, and a higher retention of sensory attributes was observed in Sample D in comparison with other samples. The sensory attributes of the samples reduced significantly (with a significance level of 0.05) over the storage period. Sample D scored the highest rating in terms of appearance ($8.92 \pm 0.07$ at Day 0 and $7.60 \pm 0.03$ at Day 50), aroma ($8.81 \pm 0.02$ at Day 0 and $7.50 \pm 0.02$ at Day 50), taste ($8.92 \pm 0.07$ at Day 0 and $7 \pm 0.03$ at Day 50), and overall acceptability ($8.90 \pm 0.05$ at Day 0 and $7.51 \pm 0.04$ at Day 50). The Hedonic scale rating of wood apple beverages for appearance ranged from $8.92 \pm 0.07$ to $7.90 \pm 0.03$ (like very much/like moderately) at the day of preparation (Day 0) and gradually decreased to the range of $7.60 \pm 0.03$ to $6.70 \pm 0.02$ (like moderately/like slightly) at the final day of preparation (Day 50). Aroma of the beverage samples was in the range of $8.81 \pm 0.02$ to $7.90 \pm 0.02$ (like very much/like moderately) at Day 0 following a gradual reduction in rating towards the end of storage. The score for aroma was in the range of $7.50 \pm 0.02$ to $6.50 \pm 0.03$ (like moderately/like slightly) at the final day of storage (Day 50). Taste of the samples were within the range of $8.92 \pm 0.07$ to $8.00 \pm 0.02$ (liked very much) at the preparation day (Day 0) and decreased to be in the range of $7.00 \pm 0.03$ to $5.50 \pm 0.02$

(like moderately—neither like or dislike) at the end of storage (Day 50). Overall acceptability of samples ranged from 8.90 ± 0.05 to 7.80 ± 0.03 (like very much/like moderately) at the day of preparation (Day 0) and followed a decreasing trend towards the end of storage. The scores for overall acceptability at the final day of storage (Day 50) were within the range of 7.51 ± 0.04 to 6.50 ± 0.02 (like moderately/like slightly). The results found in this study on the sensory attributes of wood apple beverage are in accordance with those of Sasikumar [2], Tiwari and Deen [29], Rathod et al. [14], and Chowdhury et al. [31].

The findings of this study suggest that wood apple beverage preserved with chemical preservatives retained maximum sensory attributes during storage because the 9-point Hedonic scale rating of Sample D (50 ppm KMS + 50 ppm SB + 50 ppm CA + 50 ppm AA) for sensory attributes ranged from 8.92 ± 0.07 to 7.00 ± 0.03 (Table 3) over the storage period of 50 days. It means that Sample D was either "liked very much" or "liked moderately" by the panelists over the entire storage period. This result is in accordance with that of Ayub et al. who reported that strawberry juice preserved with chemical preservatives retained maximum sensory attributes during storage [25]. It was also revealed that the use of preservatives in wood apple beverage did not have a negative influence in the opinion of consumers. This finding is supported by the reports of Akinola et al. and Sasikumar who reported that fruit beverages preserved using chemical preservatives can be acceptable to consumers [24].

**Table 3.** Sensory evaluation of samples over the storage period.

| Sensory Attributes | Sample No. | Storage Time (Days) | | | | | |
|---|---|---|---|---|---|---|---|
| | | Day 0 | Day 10 | Day 20 | Day 30 | Day 40 | Day 50 |
| Appearance | Sample A | 7.90 ± 0.03 [aF] | 7.70 ± 0.02 [aE] | 7.50 ± 0.03 [aD] | 7.20 ± 0.03 [aC] | 7.00 ± 0.02 [aB] | 6.70 ± 0.02 [aA] |
| | Sample B | 8.39 ± 0.02 [bF] | 8.10 ± 0.01 [bE] | 8.00 ± 0.03 [bD] | 7.70 ± 0.02 [bC] | 7.40 ± 0.02 [bB] | 7.00 ± 0.03 [bA] |
| | Sample C | 8.71 ± 0.02 [cF] | 8.50 ± 0.04 [cE] | 8.30 ± 0.03 [cD] | 8.00 ± 0.02 [cC] | 7.70 ± 0.01 [cB] | 7.40 ± 0.04 [cA] |
| | Sample D | 8.92 ± 0.07 [dF] | 8.80 ± 0.02 [dE] | 8.50 ± 0.03 [dD] | 8.10 ± 0.02 [dC] | 7.90 ± 0.02 [dB] | 7.60 ± 0.03 [dA] |
| Aroma | Sample A | 7.90 ± 0.02 [aF] | 7.70 ± 0.02 [aE] | 7.30 ± 0.04 [aD] | 7.00 ± 0.03 [aC] | 6.70 ± 0.02 [aB] | 6.50 ± 0.03 [aA] |
| | Sample B | 8.10 ± 0.02 [bF] | 7.80 ± 0.03 [bE] | 7.50 ± 0.03 [bD] | 7.30 ± 0.03 [bC] | 7.00 ± 0.02 [bB] | 6.70 ± 0.03 [bA] |
| | Sample C | 8.50 ± 0.02 [cF] | 8.10 ± 0.02 [cE] | 7.91 ± 0.02 [cD] | 7.50 ± 0.04 [cC] | 7.20 ± 0.03 [cB] | 7.01 ± 0.02 [cA] |
| | Sample D | 8.81 ± 0.02 [dF] | 8.60 ± 0.03 [dE] | 8.30 ± 0.02 [dD] | 8.00 ± 0.02 [dC] | 7.80 ± 0.03 [dB] | 7.50 ± 0.02 [dA] |
| Taste | Sample A | 8.00 ± 0.02 [aF] | 7.70 ± 0.02 [aE] | 7.10 ± 0.03 [aD] | 6.50 ± 0.03 [aC] | 6.00 ± 0.02 [aB] | 5.50 ± 0.02 [aA] |
| | Sample B | 8.50 ± 0.01 [bF] | 8.10 ± 0.02 [bE] | 7.90 ± 0.02 [bD] | 7.51 ± 0.02 [bC] | 6.99 ± 0.02 [bB] | 6.51 ± 0.02 [bA] |
| | Sample C | 8.80 ± 0.03 [cF] | 8.50 ± 0.02 [cE] | 8.10 ± 0.02 [cD] | 7.80 ± 0.02 [cC] | 7.30 ± 0.03 [cB] | 6.90 ± 0.03 [cA] |
| | Sample D | 8.92 ± 0.07 [dF] | 8.70 ± 0.02 [dE] | 8.31 ± 0.02 [dD] | 8.01 ± 0.02 [dC] | 7.51 ± 0.03 [dB] | 7.00 ± 0.03 [dA] |
| Overall Acceptability | Sample A | 7.80 ± 0.03 [aF] | 7.50 ± 0.02 [aE] | 7.20 ± 0.01 [aD] | 7.01 ± 0.03 [aC] | 6.70 ± 0.03 [aB] | 6.50 ± 0.02 [aA] |
| | Sample B | 8.20 ± 0.03 [bF] | 8.00 ± 0.03 [bE] | 7.60 ± 0.03 [bD] | 7.40 ± 0.03 [bC] | 7.00 ± 0.02 [bB] | 6.80 ± 0.03 [bA] |
| | Sample C | 8.60 ± 0.02 [cF] | 8.40 ± 0.03 [cE] | 8.10 ± 0.03 [cD] | 7.80 ± 0.02 [cC] | 7.50 ± 0.04 [cB] | 7.10 ± 0.03 [cA] |
| | Sample D | 8.90 ± 0.05 [dF] | 8.70 ± 0.03 [dE] | 8.40 ± 0.03 [dD] | 8.10 ± 0.02 [dC] | 7.81 ± 0.02 [dB] | 7.51 ± 0.04 [dA] |

Values represent the mean ± standard deviation of 9-point hedonic scale rating given by 150 consumers ($n = 150$); [abcdef] column—means within the different superscript letter were significantly different ($p < 0.05$); [ABCDEF] row—means within the different superscript letter were significantly different ($p < 0.05$).

## 4. Conclusions

Pasteurization at 85 °C for 10 min and a combined treatment with potassium metabisulphite, sodium benzoate, citric acid, and ascorbic acid could effectively extend the shelf life of wood apple beverages by up to 50 days. Consumption-safe parameters of wood apple beverage were properly maintained over the storage period with satisfactory consumer acceptability. Pasteurization (85 °C for 10 min) and treatment with 50 ppm potassium metabisulphite + 50 ppm sodium benzoate + 50 ppm citric acid + 50 ppm ascorbic acid was found to be more efficient than other treatments in retaining the physicochemical properties, microbial growth, and sensory attributes of wood apple beverages. Findings of this study will help interested parties to produce safe-to-consume and shelf-stable wood apple beverages on an industrial scale and promote this highly nutritious beverage commercially in the developing market of beverages.

**Author Contributions:** M.S.M. conceived, designed and carried out the experiments, handled and collected raw data, conducted statistical analysis, analyzed and interpreted the data and prepared the manuscript; W.Z. participated in manuscript preparation, facilitated data collection and reviewed the manuscript before submitting; M.B.H.S. conceptualized the project, acted as corresponding author and supervised the whole research work.

**Funding:** This research received no external funding.

**Acknowledgments:** The authors express their gratitude to the Department of Food Engineering and Tea Technology, Shahjalal University of Science and Technology, Sylhet, for continuous material support and technical assistance.

**Conflicts of Interest:** The authors declare no conflicts of interest.

## Appendix A

**Table A1.** Evaluation of physicochemical parameters of treated wood apple beverage samples during storage.

| Para Meters | Sample No. | Storage Time (Days) | | | | | |
|---|---|---|---|---|---|---|---|
| | | **Day 0** | **Day 10** | **Day 20** | **Day 30** | **Day 40** | **Day 50** |
| TSS (°Brix) | Sample A | 16.50 ± 0.03 [dA] | 16.82 ± 0.02 [cB] | 16.98 ± 0.03 [cC] | 17.40 ± 0.01 [cD] | 17.85 ± 0.03 [dE] | 18.25 ± 0.05 [cF] |
| | Sample B | 16.41 ± 0.02 [cA] | 16.51 ± 0.02 [bB] | 16.61 ± 0.02 [bC] | 16.73 ± 0.02 [bD] | 16.88 ± 0.03 [cE] | 16.91 ± 0.03 [bE] |
| | Sample C | 16.36 ± 0.03 [bA] | 16.48 ± 0.02 [bB] | 16.57 ± 0.04 [bC] | 16.62 ± 0.07 [aC] | 16.81 ± 0.03 [bD] | 16.88 ± 0.02 [bE] |
| | Sample D | 16.30 ± 0.02 [aA] | 16.40 ± 0.01 [aB] | 16.48 ± 0.02 [aC] | 16.59 ± 0.01 [aD] | 16.69 ± 0.02 [aE] | 16.78 ± 0.02 [aF] |
| pH | Sample A | 5.09 ± 0.10 [aF] | 4.91 ± 0.04 [aE] | 4.70 ± 0.02 [aD] | 4.44 ± 0.05 [aC] | 4.30 ± 0.05 [aB] | 4.10 ± 0.02 [aA] |
| | Sample B | 5.20 ± 0.02 [bF] | 5.15 ± 0.02 [bE] | 5.09 ± 0.02 [bD] | 5.00 ± 0.02 [bC] | 4.95 ± 0.02 [bB] | 4.73 ± 0.02 [bA] |
| | Sample C | 5.30 ± 0.02 [bcF] | 5.22 ± 0.02 [cE] | 5.15 ± 0.02 [cD] | 5.08 ± 0.02 [cC] | 4.98 ± 0.02 [bcB] | 4.78 ± 0.02 [cA] |
| | Sample D | 5.43 ± 0.01 [cF] | 5.34 ± 0.02 [dE] | 5.28 ± 0.02 [dD] | 5.15 ± 0.01 [dC] | 5.08 ± 0.02 [cB] | 4.97 ± 0.03 [dA] |
| Acidity (%) | Sample A | 0.43 ± 0.02 [aF] | 0.37 ± 0.01 [aE] | 0.26 ± 0.02 [aD] | 0.17 ± 0.02 [aC] | 0.12 ± 0.01 [aB] | 0.08 ± 0.01 [aA] |
| | Sample B | 0.57 ± 0.02 [bE] | 0.52 ± 0.02 [bD] | 0.47 ± 0.01 [bC] | 0.41 ± 0.01 [bB] | 0.39 ± 0.01 [bB] | 0.34 ± 0.01 [bA] |
| | Sample C | 0.60 ± 0.02 [cF] | 0.56 ± 0.01 [cE] | 0.53 ± 0.01 [cD] | 0.49 ± 0.01 [cC] | 0.45 ± 0.006 [cB] | 0.36 ± 0.01 [cA] |
| | Sample D | 0.69 ± 0.01 [dF] | 0.63 ± 0.006 [dE] | 0.58 ± 0.006 [dD] | 0.53 ± 0.01 [dC] | 0.49 ± 0.01 [dB] | 0.43 ± 0.01 [dA] |
| Ascorbic Acid Content (mg/100mL) | Sample A | 4.23 ± 0.02 [aF] | 3.41 ± 0.02 [aE] | 2.74 ± 0.04 [aD] | 2.11 ± 0.04 [aC] | 1.44 ± 0.05 [aB] | 1.01 ± 0.07 [aA] |
| | Sample B | 4.40 ± 0.02 [bF] | 4.10 ± 0.02 [bE] | 3.80 ± 0.01 [bD] | 3.31 ± 0.01 [bC] | 2.99 ± 0.02 [bB] | 2.77 ± 0.01 [bA] |
| | Sample C | 4.50 ± 0.02 [cF] | 4.17 ± 0.01 [cE] | 3.89 ± 0.01 [cD] | 3.49 ± 0.01 [cC] | 3.10 ± 0.02 [cB] | 2.84 ± 0.01 [cA] |
| | Sample D | 4.65 ± 0.02 [dF] | 4.25 ± 0.01 [dE] | 4.00 ± 0.02 [dD] | 3.70 ± 0.01 [dC] | 3.39 ± 0.02 [dB] | 3.01 ± 0.02 [dA] |

Values represent the mean ± standard deviation of 3 replicates ($n$ = 3); [abcdef] column—means within the different superscript letter were significantly different ($p < 0.05$); [ABCDEF] row—means within the different superscript letter were significantly different ($p < 0.05$).

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
