# Peer review of "Shelf-Life Extension of Wood Apple Beverages Maintaining Consumption-Safe Parameters and Sensory Qualities"

_beverages, doi:10.3390/beverages5010025_

Reviewer 1 Report

To Authors

The manuscript (ref. Beverages-419066) titled “Shelf-life Extension of Wood Apple Beverage Maintaining Consumption Safety Parameters and Sensory Qualities” by Moazzem & Sikder submitted to Beverages, aimed at prolonging shelf-life of wood apple beverage while ensuring safety for human consumption and keeping its sensory quality by adding potassium metabisulphite, citric acid, ascorbic acid, and sodium benzoate to pasteurized samples. Authors “were able to extend the shelf-life of wood apple beverage up to 50 days satisfactorily from its natural shelf-life of 8-12 hours.” 

The paper’s English requires some revision in order to more clearly convey Authors ideas about the relatively strait-forward experiment carried out about an important(?) beverage.

In the Introduction, revision of text and updating of few refs to newer studies would help better contextualize the study and its importance/relevance. Is wood apple juice an importante beverage? if so, provide indo/data to back that affirmation. If not, emphasize its potential to become a more popular beverage eventually of commercial interest. Authors should consider including a sentence or two about the legal context/regulations that apply to use of food additives (such as those in force in Bangladesh, India or Asia or even in the European Union or in the US).

A few informations for completion of the description of the methodology are needed in the Material and Methods section. Attention should be given to the use of units herein but also throughout the text: use g for gram and grams, L for liter (Authors use mL for mililiter). Also, use pH instead of p^H (^ superscipt). Adding methodological details (e.g. in subsection 2.7) would improve this section. In terms of the experimental design followed (L. 99-105)., why didn't the Authors consider a 2-level factorial experimental design to plan their experiment? In the subsection dealing with sensory analysis (L. 122-126), Authors are encouraged to provide more info about the panel.

Authors should consider starting the Results and Discussion section (as well as the subsections) with their results (move L. 146-151 to L.142). Also, using words such as "better" or "best" or "worst" sounds judgmental, subjective. Consider rewording L. 146-151 and elsewhere in text. Subsections 3.1 thru 3.4 could be merged into one subsection dealing with physicochemical parameters. Seemingly the refs cited in this section are relatively dated (1970s-1990s); Authors are encouraged to cite more recent refs. In Figs 2-5 label the X-axis “Storage time (days)”. Consider merging Figs 2-5 into one composite figure i.e. a panel of 2x2 plots. Check the captions, the legends and headings of Tables 1-4. Besides, these tables seem to duplicate (in more detail) the data presented in Figs 2-5. The subsubsections in section 3.5 Microbial Analysis of Samples as well as the sub subsubsections 3.6.1-3.6.4 could be merged and thus the observable repetition in text could be avoided. Refs cited in the introductory sections are more recent than those cited in the Results and Discussion section. Authors are encouraged to find, read and complement the latter with more recent published papers (if available). 

These and more specific comments and suggestions are made directly to the submitted PDF using the tools available in Adobe Acrobat Reader DC. Surely Authors will be able to use them to improve the manuscript.

Author Response

1. We thank the reviewer for his/her insightful comments about our manuscript. The reviewer reported that “Moderate English changes required” in his/her review report. We have revised the English language and style in our revised manuscript in order to more clearly convey our ideas.

2. The reviewer has asked “Is wood apple juice an important beverage?”. In our investigation, we found that wood apple beverage holds tremendous potential to become a more popular beverage eventually of commercial interest and we have emphasized this in our revised manuscript. We have also provided recent information and published papers to support our findings.

3. In the Introduction, we have revised the text and updated many references to newer studies according to the comment of the reviewer. We have included the legal context / regulations of Bangladesh, India, USA and FAO for the use of food additives. Information for the completion of the description of the methodology have been updated in the “Material and Methods” section. Attention has been given to the use of units (such as ml, pH etc.) throughout the revised manuscript according to the suggestion of the reviewer.

4. Methodological details have been added in the microbial analysis section (section 2.7 of the previous version of the manuscript, which has been labelled as section 2.8 in the revised version) to improve this section according to the suggestion of the reviewer.

5. The reviewer has asked, “Why didn't the Authors consider a 2-level factorial experimental design to plan their experiment?” We used the conventional method to design our experiment.

6. In the subsection dealing with sensory analysis, we have provided more info about the panel according to the suggestion of the reviewer.

7. We have started the “Results and Discussion” section (as well as the subsections) with their results as per the suggestion of the reviewer. We have avoided the use of “best" or "worst” for the interpretation of results but used the word “better” to compare the differences of results among the samples.

8. We reworded line 146-151 elsewhere in text as per the suggestion of the reviewer. Subsections 3.1 through 3.4 have been merged into one subsection dealing with physicochemical parameters as per the suggestion of the reviewer. We have cited more recent references (2008-2018) in this section. We have labelled the X-axis “Storage time (days)” in Figure 2-5 as per the suggestion of the reviewer.

9. The reviewer has suggested that “Consider merging figure 2-5 into one composite figure i.e. a panel of 2x2 plots.” We think that it will be hard to interpret results more accurately if they are merged in one composite figure. So, we did not merge the figures.

10. The captions, the legends and headings of Tables 1-4 were checked and we have merged these tables and we have placed the merged table in the “Appendix” section of our revised manuscript.

11. The sub-subsections in section 3.5 (microbial Analysis of samples) as well as the sub sub-subsections 3.6.1-3.6.4 have been merged according to the suggestion of the reviewer. We have updated our revised manuscript with more recent published papers according to the suggestion of the reviewer.

Reviewer 2 Report

The manuscript titled " Shelf-life Extension of Wood Apple Beverage Maintaining Consumption Safety Parameters and Sensory Qualities” presented the study about extending the shelf-life as well as sensory quality of Wood Apple beverage up to at least 50 days by adding different preservatives.

One of the weakness part of the manuscript are referances. There were cited 34 position, nevertheless only 11 of them are from last 10 years, the rest position are from: 1958 – 2005

The manuscript needs to be improved and same correction and explanation have to be done  according to the reviewer comments below:

Line 31: remove „as such”

Line 37: How fruits can be used as a liver?

Line 41: what the „gm” abbreviation means?
there are no such abbreviation of units

Line 41-44: Wood Apple fruits are richest source of Vitamin B2 as it contains 1191 mg/ 44 100g riboflavin” in the sentence above is written that that wood apple contain 0,2 mg vitamin B2 – it’s confusing.

Line 45: Before given an example of use preservatives in juice, specify and explain why this compounds (preservatives) are so necessary in such products.

Line 73-76: why wood apple once is written with big letter and other time with small letter? What is the difference? Unify the spelling throughout the manuscript.

Line 90, 91, 92, 93,  what the „gm” abbreviation means?
there are no such abbreviation of units

Line 100, 101, 102, and 104: add “as” befor Sample B, Sample C and sample D

Figure 1. Step 7 what the „gm” abbreviation means?
there are no such abbreviation of units

Line 110-114: why “H” in pH is written with upper index?

Line 119: The total plate count is the enumeration of aerobic, mesophillic organisms that grow in aerobic conditions under moderate temperatures of 20-45°C. Specify briefly the methods used to enumerating total plate count. Also, described what kind of bacteria, yeast and molds did authors analysis, due to the fact that the beverage was pasteurized the main group of microorganism that could contaminate this product are for example spore-forming thermotolerant bacteria but also same of yeast and molds.Additionaly why the beverage was not check against main hazard microorganism like E. coli and also pathogenic bacteria?

Line 128: why “H” in pH is written with upper index?

Line 133: why “H” in pH is written with upper index?

Line155-156:
TSS results are placed both on the figure 2 and in the table 1. The reviewer's opinion should not repeat the same results. TSS results should be presented only once in a figure1 or in a table 1.

Line 160-170: why “H” in pH is written with upper index?

Also, pH results are placed both on the figure 3 and in the table 2. The reviewer's opinion should not repeat the same results. pH results should be presented only once in a figure 3 or in a table 2.

Line 165: Instead of “However, I found…” should be “However, it was found that …”

Line 168: “Figure 3 shown that…”

Line 178: sentence should be written impersonally,  we do not use it in scientific works: he did, I did - rewrite the sentenceLine 185: remove double bracket 
Line 190-191: Acidity results are placed both on the figure 4 and in the table 3. The reviewer's opinion should not repeat the same results. Acidity results should be presented only once in a figure 4 or in a table 3. 
Line 196-200: the discussion based on articles from 1970, 1988 and 1961,
if there are any newer, more current articles on the basis of which the discussion can be held? 

Line 199: remove dot after squash

Line 212-213: Ascorbic acid content results are placed both on the figure 5 and in the table 4. The reviewer's opinion should not repeat the same results. Ascorbic acid content results should be presented only once in a figure 5 or in a table 4.

Line 223-224 and 231-232: why in untreated control samples the presence of bacteria, yeast and mold are so high? The products was pasteurized and a larger number of microorganisms should be destroyed during this process. 
Line 232: incorrect description of bacteria number instead of 109x105 cfu/mL should be 1,09x107 cfu/mL; instead of 243x105 cfu/mL should be 2,43x107 cfu/mL, check and correct the description of bacteria, yeast and molds in whole manuscript and in tables 5 and 6. 
Table 5: what the „nil” abbreviation means? Should be explain under the table. 
Line 233-234: “The presence of bacterial load was noticed at the end period of storage, notably at Day 40 (3×105 cfu /ml)” according to the information in table 5, after 40 days of storage the bacteria count was detected under 3×105 and it should be notice in the text. This comment is also applies to the line: 235, 247 and 248. 
Table 5: what is the count of bacteria in samples B after 50 day, if the count is under 2x105, why the authors did not determine exactly the number of bacteria conatmination.
This comment also applies to the table 6, why the exactly number of yeast and mold are not detected in samples B and samples C after 50 days. 
Line 281-282: “Adjustment of acidity within 282 the range of .2-.35…”-confusing what does .2-.35 mean? 
Sensory analysis: why this product on the 9-point Hedonic Rating Scale before storage on the first day are so low below 5,8 to 7,5; panellists rated this product at the very beginning as “like moderately to like slightly”. In this situation, with such assessments, whether will the product be accepted by consumers. Before storage the product should obtain the highest sensory evaluation. 

Author Response

Dear Sir, 

Please find the reply with the attached file.

Reviewer 3 Report

The coments are in the manuscript pfd

Author Response

1. We thank the reviewer for his/her insightful comments about our manuscript. The reviewer reported that “Moderate English changes required” in his/her review report. We have revised the English language and style in our revised manuscript in order to more clearly convey our ideas.

2. The reviewer has asked “Is wood apple juice an important beverage?”. In our investigation, we found that wood apple beverage holds tremendous potential to become a more popular beverage eventually of commercial interest and we have emphasized this in our revised manuscript. We have also provided recent information and published papers to support our findings.

3. In the Introduction, we have revised the text and updated many references to newer studies according to the comment of the reviewer. We have included the legal context / regulations of Bangladesh, India, USA and FAO for the use of food additives. Information for the completion of the description of the methodology have been updated in the “Material and Methods” section. Attention has been given to the use of units (such as ml, pH etc.) throughout the revised manuscript according to the suggestion of the reviewer.

4. Methodological details have been added in the microbial analysis section (section 2.7 of the previous version of the manuscript, which has been labelled as section 2.8 in the revised version) to improve this section according to the suggestion of the reviewer.

5. The reviewer has asked, “Why didn't the Authors consider a 2-level factorial experimental design to plan their experiment?” We used the conventional method to design our experiment.

6. In the subsection dealing with sensory analysis, we have provided more info about the panel according to the suggestion of the reviewer.

7. We have started the “Results and Discussion” section (as well as the subsections) with their results as per the suggestion of the reviewer. We have avoided the use of “best" or "worst” for the interpretation of results but used the word “better” to compare the differences of results among the samples.

8. We reworded line 146-151 elsewhere in text as per the suggestion of the reviewer. Subsections 3.1 through 3.4 have been merged into one subsection dealing with physicochemical parameters as per the suggestion of the reviewer. We have cited more recent references (2008-2018) in this section. We have labelled the X-axis “Storage time (days)” in Figure 2-5 as per the suggestion of the reviewer.

9. The reviewer has suggested that “Consider merging figure 2-5 into one composite figure i.e. a panel of 2x2 plots.” We think that it will be hard to interpret results more accurately if they are merged in one composite figure. So, we did not merge the figures.

10. The captions, the legends and headings of Tables 1-4 were checked and we have merged these tables and we have placed the merged table in the “Appendix” section of our revised manuscript.

11. The sub-subsections in section 3.5 (microbial Analysis of samples) as well as the sub sub-subsections 3.6.1-3.6.4 have been merged according to the suggestion of the reviewer. We have updated our revised manuscript with more recent published papers according to the suggestion of the reviewer.

Round  2

Reviewer 1 Report

To the Authors

In the revised manuscript (beverages-419066-v2), Authors addressed several of the questions raised in the original manuscript including the moderate changes to the text’ English required. A few, relatively minor issues remain, namely:

reference to results/values in lines 17-18, 

“excessive” instead in line 66, 

citation format in lines 84, 90, 92, 97, 100, 102, 105, 106, 108, 111, 113, 115, 243-246, 251-253, 260-262, 269-271, 297-299, 323-324, 329

clarification of text (mesh size in mm?) in line 150

L instead of “liter” in lines 155-156

Insert Table 1 was ‘good choice’

Changes to text (line 171) compared to original ms

Suggestions to text in lines 176-177, 180, 208, 237, 249, 267, 268, 276, 280, 296, 303, 305-307, 309, 323, 326, 338

Text in lines 231-232  (and the text above in subsection 2.9) indicates that the Authors carry out supplementary experimental work? 

Add details about p-value and stats tests when stating ’significance’ as in line 278

Are the initial microorganisms loads found safe i.e when compared to the recommendations by ICMSF (vd. http://www.icmsf.org)

When referring average values (e.g. lines 309-) adding a measure of variation/dispersion of original values, such as std var. is expected.

Complete ‘idea’ in line 327.

In Conclusion (line 336), Authors state “Pasteurization at 85ᵒC for 20 minutes and…” but previously throughout the text (e.g. in line 159), 10 minutes is stated as the pasteurization time? 

Check (and standardize) the references e.g. full or abbreviated journal titles?, in the final list.

These are also annotated in the PDF of the revised version of the manuscript using the commenting tools in Adobe Acrobat Reader DC. Authors will be able to use them.

Author Response

Dear Sir, 

please find our detailed response in the attached file.

Reviewer 2 Report

Minor revision should be done before final publication:

Line 14: Controls – write with lowercase letters

Line 49: People – write with lowercase letters

Line 66: Excessive – write with lowercase letters

Line: 282: According to the methods used the number of mesophilic bacteria was determined. Beside what does mean thermotrophic bacteria-explain and give same example of this bacteria?

Line 480: deleted from the manuscript “names of proposed reviewers”

Author Response

Dear Sir,

Please find our detailed response in the attached file.
